# Prevalence and associated factors of suicidal ideation and attempt among undergraduate medical students of Haramaya University, Ethiopia. A cross sectional study

Henock Asfaw[1]ᵒ*, Niguse Yigzaw[2]ᵒ, Zegeye Yohannis[3]ᵒ, Gelana Fekadu[4]ᵒ, Yadeta Alemayehu[5]ᵒ

1 Department of Psychiatry, School of Nursing and Midwifery, College of Health and Medical Science, Haramaya University, Harar, Ethiopia, 2 Department of Psychiatry, School of Medicine, College of Health and Medical Science, University of Gonder, Gonder, Ethiopia, 3 Department of Psychiatry, Amanuel Mental Specialized Hospital, Addis Ababa, Ethiopia, 4 Department of Nursing, School of Nursing and Midwifery, College of Health and Medical Science, Haramaya University, Harar, Ethiopia, 5 Department of Nursing, School of Nursing and Midwifery, College of Health and Medicalc Science, University of Mettu, Mettu, Ethiopia

ᵒ These authors contributed equally to this work.
* yaredasfaw33@gmail.com

**Data Availability Statement:** All relevant data are within the paper and its Supporting Information files.

## Abstract

### Introduction

Suicide is a disastrous act which has a significant effect on the global burden of disease, contributing yearly to 1.4% of the total burden with the foremost role played by a people aged between 15 and 35 years. Medical students are one of the high-risk groups for suicide. This increased risk may begin during medical school and leads to premature death. But, there is a paucity of epidemiologically reliable data on the issue. Therefore, the current study was aimed to assess the prevalence and associated factors of suicidal ideations and attempt among undergraduate medical students of Haramaya University.

### Method

An institutional based cross-sectional study was conducted from May 13 to June 12, 2019 at College of Health and Medical Science, Haramaya University. Stratified sampling technique was used and a total of 757 participants were selected by using simple random sampling technique. Data were collected using a self-administered questionnaire. Suicidal ideation and attempt were assessed by using suicidal module of world mental health survey initiative version of the World Health Organization, composite international diagnostic interview. Data were analyzed using Statistical Package for Social Science Version 20. Descriptive results were presented by tables and graphs. Bivariate and multivariate logistic regression analyses were done to identify factors associated with suicidal ideation and attempt. P-values less than 0.05 were considered statistically significant and the strength of association was presented by an adjusted odds ratio with 95% confidence interval.

**Funding:** Funding for this study was provided by a joint program from University of Gondar and Amanuel mental specialized hospital.

**Competing interests:** The authors have declared that no competing interests exist.

**Abbreviations:** AOR, Adjusted Odds Ratio; ASSIST, Alcohol Smoking and Substance Involvement Screening Test; CI, Confidence Interval; CIDI, Composite International Diagnostic Interview; COR, Crude Odds Ratio; DASS, Depression, Anxiety, Stress Scale; SD, Standard Deviation.

## Result

The study showed that the prevalence of suicidal ideation and attempt were 23.7% (95%CI, 20.5–26.8) and 3.9% (95%CI, 2.6–5.5), respectively. Cumulative grade point average (AOR = 0.30, 95% CI: 0.18–0.49), current alcohol use (AOR = 2.26, 95%CI: 1.45–3.55), depression (AOR = 3.58, 95%CI: 2.23–5.76), anxiety (AOR = 3, 95%CI: 1.88–4.77), and poor social support (AOR = 2.57, 95%CI: 1.41–4.68) were the factors statistically associated with the suicidal ideation. Depression (AOR = 5.4, 95%CI: 1.45–20.14) and anxiety (AOR = 3.19, 95%CI: 1.01–10.18) were associated with the suicidal attempts.

## Conclusion

This study showed that the high prevalence of suicidal ideation and attempt as compared to the prevalence of suicidal behavior among other university students who were studying in other fields. Cumulative Grade Point Average, current alcohol use, depression, anxiety and poor social support were the factors statistically associated with the suicidal ideation. Depression and anxiety were the ones associated with the suicidal attempt. Early screening, detection and management of suicidal behavior and associated mental health problems were recommended for undergraduate medical students.

## Introduction

According to the World Health Organization (WHO) estimate, annually around 800,000 people die due to suicide. This corresponds to an age-standardized suicide rate of 11.5 per 100,000 people and a figure equivalent to someone dying of suicide every 40 seconds. Also the rate has increased by 60% over the last 50 years worldwide [1]. It is the second leading cause of death among adolescents and young adults, accounting for 8.64% [2, 3].

Literatures revealed that one in four people know someone who has taken their own life and that one suicidal death leaves six or more suicide survivors [4]. The survivors represent the largest mental health causality related suicide [5]. It takes more time to the significant others to heal from the pain of losing loved one by suicide than the other disease. The bereavement process after the unsuccessful attempt may also end with suicide [6].

Medical school is one of the most stressful environment. Medical students are prone to mental distress due to a problem with adjustment to the medical school environment, ethical issues, and witnessing death and human suffering [7]. These could lead to poor academic performance, substance abuse, academic dishonesty, and suicide [8].

Death which is caused by suicide is a major work related hazard among medical professionals, and the risk may begin during medical school [9]. Studies have suggested that the suicide rate among medical students is higher than age-matched general population, and also it is reported to be the second most common cause of mortality among medical students in USA [10, 11]. The twelve months and lifetime prevalence of suicidal ideation among medical students ranges from 7% to 35.6% and 2.9% to 53.6%, respectively [12].

In low and middle income countries, the rate of suicidal ideation among medical students is higher than the general population [10, 13]. Different study results revealed that the rate of twelve months suicidal ideation and attempt among medical students in Austria, Turkey, Pakistan, and China were 11.3% and 0.3%; 12% and 2.1%; 35.6% and 4.8%, and 8.2% and 4.3% respectively [13–15].

Higher suicidal ideations and attempts were reported among medical students in African, reaching 32.3% and 6.9% among South African [10] and 12.75% among Egypt students [16]. However, there has been no study report of suicidal ideation and attempt among medical students in Ethiopia, but the magnitude of suicidal behavior among general population ranges from 0.9% to 60% for suicidal ideation [17] and 3.8% to 27% for suicidal attempt [18–21].

Suicide prevention should focus on early identification of the risk factors and intervention for high risk students. Medical students seem tremendously vulnerable to suicidal ideation because of overwhelming stress put by academic and non-academic issues [22, 23]. The most frequently reported factors correlated with suicidal behavior are being male, psychological distress [15], depression [24], dissatisfaction with academic performance, feeling neglected by parents [25, 26], substance abuse, psychiatric disorders [13], drug use like opioid [27], depressive symptoms [10], first year and pre-clinical phase, homesick [16], alteration in thyroid, and prolactin hormone [28]. The loss of loved one by suicide and emotional turmoil related to bereavement also linked with increase in suicidal behavior [6].

There is a paucity of epidemiologically reliable data on suicidal ideation and attempt among medical students in Ethiopia. Therefore, the current study aimed to assess the prevalence and the associated factors of suicidal ideations and attempts among undergraduate medical students of Haramaya University, Ethiopia.

## Methods and materials

### Study period and setting

The study was carried out from May 13 to June 12, 2019 in College of Health and Medical Science, Haramaya University, Ethiopia. Haramaya University is located in eastern part of the country, at a distance of 510 km from the capital, Addis Ababa. The university has nine colleges in two campuses. One is Harar campus (college of health and medical science), which is located 17 km from the main campus and includes both medical and health science programs. The School of Medicine was emerged as an academic constituent in Haramaya University, College of Health and Medical Science in 2007 and it enrolled its first batch in 2008. Currently the school has 1315 students learning in its undergraduate medical program.

### Study design and population

An institutional based cross sectional study design was used. All randomly selected undergraduate medical students in the school were included in the study and students transferred from other universities and stayed for less than six month in the university were excluded.

### Sample size and sampling procedure

Single population proportion formula was used to calculate the sample size by considering the assumptions: Z = standard normal distribution (Z = 1.96), with confidence interval of 95%, P = the prevalence of suicidal ideation among Haramaya University students 20.2% [29], d = margin of error = 3%. Then, adding 10% (688 x 0.1 = 69) of non-respondent, the total sample size was 757.

Concerning the sampling procedure, first the students were stratified based on their academic year, and then the total sample size was allocated proportionally to each academic year. Finally, a simple random sampling technique was used to select study participants from each academic year (Fig 1).

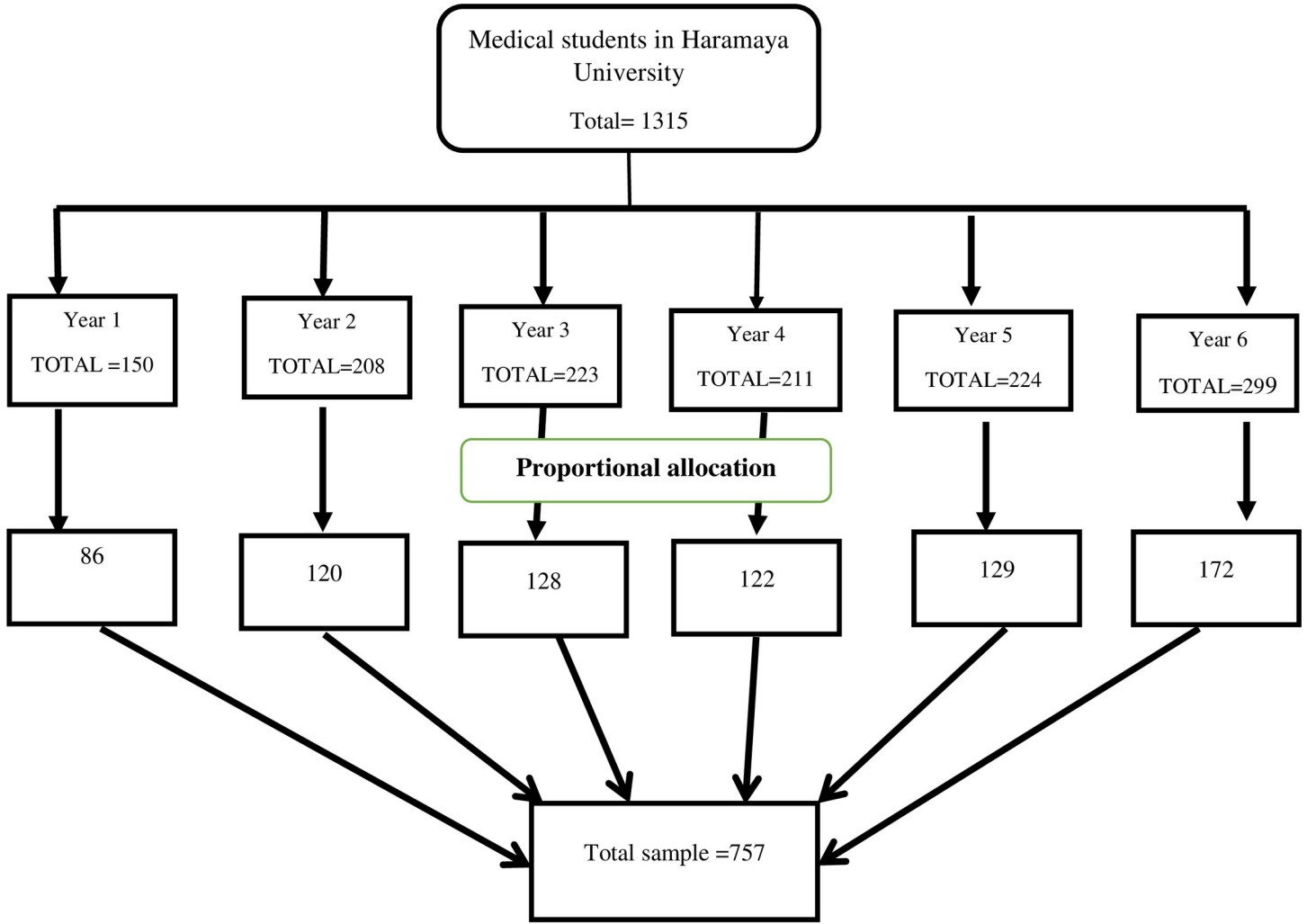

**Fig 1. Schematic presentation of sampling procedure for the assessment of prevalence and associated factors of suicidal ideation and attempt among undergraduate medical students of Haramaya University, Harar, Ethiopia, 2019.**

## Data collection tool and procedure

The data were collected by using a self-administered questionnaire which had items on socio-demographic characteristics like, age, sex, marital status, living conditions, and religion. Family factors like family history of suicidal attempt, of committed suicide, and of mental illness. ASSIST(Alcohol, Smoking and Substance Involvement Screening Test), which is developed by WHO [30], was used to collect data related to psychoactive substances.

DASS (depression, anxiety, and stress scale) was used which has twenty one items. It was Lakert scale ranges from zero to three. A sum score of ten and above was taken as indicator of the presence of depression, eight and above of anxiety, and fifteen and above of stress [31]. Data on social support were collected by Oslo-three item social support scale. A sum score of 3 to 14 was labeled as having poor support, 3 to 8 moderate support, 9–11 and strong support 12–14 [32].

The outcome variables (suicidal ideation and attempt) were assessed by "Yes" or "No" questions coded as 1 for "Yes" and 0 for "No". The questions were adapted from module of World mental health survey initiative version of the WHO, CIDI (Composite International

Diagnostic Interview). It has been used and validated in Ethiopia both at clinical and community settings [33].

The data were collected by four BSc degree psychiatric nurses and supervised by one master's degree holder in psychiatry. The questionnaire was adapted and administrated in English.

### Data quality control

A pre-test was done on 38 medical students (5%of the total sample) in Dire Dawa University and necessary amendments were made. During data collection process, the questionnaire was checked for its completeness on daily basis by the supervisors and the investigators.

### Data processing and analysis

The data were checked and entered Epi-Data Version 3.1 and exported to SPSS (Statistical Package for Social Science) Version 20 for analysis. The socio-demographic characteristics and other factors were analyzed by descriptive statistics (percentage, mean and standard deviations). Bivariate and multivariate logistic regression analyses were performed to identify the factors associated with the outcome variables. All the variables with a $p$-value less than 0.20 in the bivariate analysis were entered the multivariate logistic regression model. Here, a factor with a $p$-value of $< 0.05$ was considered statistically significant, and the adjusted odds ratio (AOR) with 95% confidence interval (CI) was used.

### Ethical considerations

Ethical approval and clearance was obtained from a joint ethical review committee of the University of Gondar, College of medicine and health sciences and Amanuel mental specialized hospital. Formal letter was obtained from Amanuel mental specialized hospital and submitted to Haramaya University, College of Health and Medical Sciences for administrative approval prior to the data collection. Informed written consent was obtained from the students prior to the data collection. Those who did not want to take part in the study were allowed not to participate or to withdraw from the study at any time they want.

## Results

### Socio-demographic characteristics of the respondents

Seven hundred ten medical students were participated in the study, with the response rate of 94%. The mean age of the respondents was 22.71(±2.62SD) years, with age ranging from 18–32 years. Many of them were male, 489 (68.9%), 554(78%) were living in dormitory, 355 (50%) were Orthodox by religion, and 652 (91.8%) were single. The mean grade of the participants were 2.99 (±0. 43SD), ranging from 1.83–3.87 (Table 1).

### Prevalence of suicidal ideation, attempt and psychosocial characteristics

The twelve month prevalence of suicidal ideation and of attempt among the participants were 23.7% (95%CI: 20.5–26.8) and 3.9% (95% CI: 2.6–5.5), respectively. Twenty eight (62.2%) respondents had attempted suicide once, 15(33.3%) had done it twice, and 2(4.4%) had done so more than two times. The most commonly used method of an attempt was poisoning 22 (48.9%), followed by hanging 19(42.2%). Among the respondents who had attempted suicide, 22(48.9%) were serious or determined to kill themselves, whereas 21(46.7%) merely tried. Of the study participants, 249 (35.1%) had had poor social support, 311(43.8%) had moderate, and 150(21.1%), enjoyed strong social support. About 334(47%) were anxious, 324(45.6%)

**Table 1. Socio-demographic characteristics of undergraduate medical students of Haramaya University, Harar, Ethiopia, 2019 (n = 710).**

| Variable | Category | Frequency | Percentage |
|---|---|---|---|
| Age | Mean | Standard deviation | |
| | 22.71 | ±2.62 | |
| Sex | Male | 489 | 68.9 |
| | Female | 221 | 31.1 |
| Marital status | Single | 652 | 91.8 |
| | Married | 52 | 7.3 |
| | Others* | 6 | 0.8 |
| Living situation | In dormitory | 554 | 78.0 |
| | Rented house | 81 | 11.4 |
| | With parents | 75 | 10.6 |
| Religion | Orthodox | 355 | 50 |
| | Muslim | 182 | 25.6 |
| | Protestant | 150 | 21.1 |
| | Others** | 23 | 3.2 |
| Educational level | First year | 85 | 12.0 |
| | Second year | 116 | 16.3 |
| | Third year | 118 | 16.6 |
| | Fourth year | 112 | 15.8 |
| | Fifth year | 121 | 17.0 |
| | Sixth year | 158 | 22.3 |

Others* = divorced, separated and widowed

Others** = Wakefata, Atheist, Catholic, Jehovah witness and Apostle.

were depressed and 322(45.4%) were stressed and also 52(7.3%) and 32 (4.5%) of the students reported family history of attempted and committed suicide, respectively (Table 2).

## Substance use

Few of the respondents, 198(27.9%), were alcohol users, 139(19.6%) were khat chewers and 54 (7.6%) smoked cigarette in the past three months (Fig 2).

## Factors associated with suicidal ideation

In this study, the odds of having suicidal ideation among alcohol user-participants was about 2.26 [AOR = 2.26, 95% CI: (1.45–3.55)] times higher as compared with non-users. For one unit increase in grade point average, the odds of having suicidal ideation decreased by 70%, [AOR = 0.30, 95% CI: (0.18–0.49)]. The odds of having suicidal ideation and attempt among the participants with depression was about 3.58[AOR = 3.58, 95% CI: (2.23–5.76)] and 5.4[AOR = 5.40, 95%CI: (1.45–20.14)] times higher than among their counterparts. The odds of having suicidal ideation and attempt among participants with anxiety was about three [AOR = 3, 95% CI: (1.88–4.77)] and 3.19 [AOR = 3.19, 95%CI: (1.01–10.18)] times higher than their counterparts respectively. The participants with poor social support were 2.57[AOR = 2.57, 95% CI: (1.41–4.68)] times more likely to have suicidal ideation as compared to those with strong social support (Table 3).

## Discussion

The current study identified that the twelve month prevalence of suicidal ideation among the undergraduate students was 23.7% (95% CI: 20.5–26.8), and this is in line with a study result

**Table 2. Suicidal ideation, attempt and psychosocial characteristics of undergraduate medical students of Haramaya University, Harar, Ethiopia, 2019 (n = 710).**

| Variable | Category | Frequency | Percentage |
|---|---|---|---|
| Suicidal ideation in 12 months | Yes | 168 | 23.7 |
| | No | 542 | 76.3 |
| Ever suicidal ideation | Yes | 208 | 29.3 |
| | No | 502 | 70.7 |
| Ever plan of suicide | Yes | 28 | 3.9 |
| | No | 682 | 96.1 |
| Suicidal attempt in 12 months | Yes | 28 | 3.9 |
| | No | 682 | 96.1 |
| Ever suicidal attempt | Yes | 45 | 6.3 |
| | No | 665 | 93.7 |
| Frequency of suicide attempt | Once | 28 | 62.2 |
| | Twice | 15 | 33.3 |
| | More than twice | 2 | 4.4 |
| Reason for suicidal attempt | Family conflict | 14 | 31.1 |
| | Death of family | 16 | 35.6 |
| | Financial loss | 10 | 22.2 |
| | Mental illness | 1 | 2.2 |
| | Physical illness | 2 | 4.4 |
| | Others* | 2 | 4.4 |
| Severity related to attempt | Seriously attempted | 22 | 48.9 |
| | In effective method | 21 | 46.7 |
| | To seek help | 2 | 4.4 |
| Methods of attempt | Hanging | 19 | 42.2 |
| | Poisoning | 22 | 48.9 |
| | Sharp tools | 2 | 4.4 |
| | Others** | 2 | 4.4 |
| Depressive | Yes | 324 | 45.6 |
| | No | 386 | 54.4 |
| Anxiety | Yes | 334 | 47.0 |
| | No | 376 | 53.0 |
| Stress | Yes | 322 | 45.4 |
| | No | 388 | 54.6 |
| Family History of suicidal attempt | Yes | 52 | 7.3 |
| | No | 658 | 92.7 |
| Family History of committed suicide | Yes | 32 | 4.5 |
| | No | 678 | 95.5 |
| Family History of mental illness | Yes | 68 | 9.6 |
| | No | 642 | 90.4 |

Others*: relationship problem

Others**: drug overdose.

in Pakistan, which accounted 24.6% [34]. However, the result of this study is lower than the ones reported from South Africa (32.3%) [10] and India (53.6%) [35]. The possible reason for discrepancy might be lower sample size in South Africa and India. On the other hand, the prevalence we found is higher than those reported from studies done in USA (9.4%) [36], Germany (14.7%) [37], China (7.5%) [24], Nepal (10.5%) [26], Austria (11.3%) [14], Turkey (12%) [14], and Egypt (12.75%) [16]. The possible reason for dissimilitude might be differences in

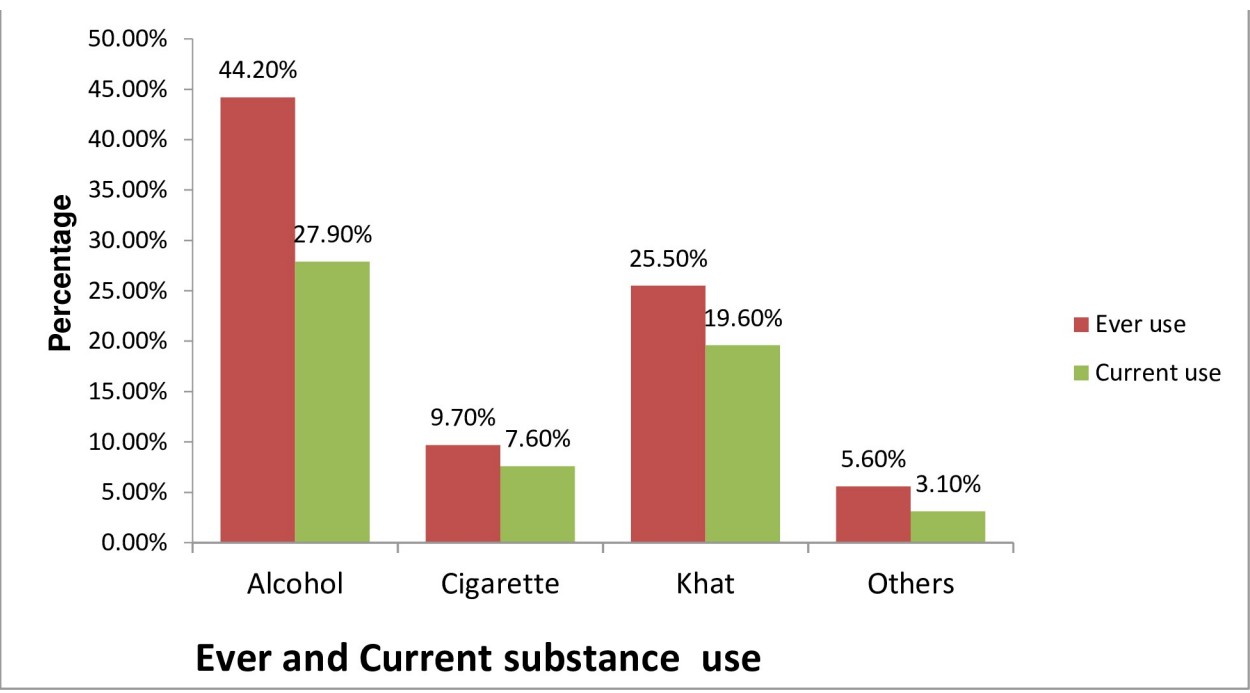

**Fig 2. Ever and current substance use among undergraduate medical students of Haramaya University, Harar, Ethiopia, 2019 (n = 710).** Others: Cannabis and Shisha.

sample size, study design, and tools used. The studies in Germany and China were prospective cohort studies and used patient health questionnaire items. In Egypt, Beck suicidal questionnaire, which assess suicidal ideation in the past two weeks, was used. Socio-economic and cultural differences might also be another possible reasons for the disagreement [38].

The magnitude of suicidal attempt in this study was 3.9% (95% CI: 2.6–5.5). This is in agreement with the ones found by similar studies conducted in Colombia (5%) [39], China (4.3%) [15], and Pakistan (4.8%) [13]. However, it is higher than those reported from Portugal (0.7%) [40], Austria (0.3%) [14], Turkey) (2.1% [14], and United Arab Emirates (1.8%) [41]. The difference might be due to the fact that, unlike our study, which included students from each academic year (year 1 to 6), in the study conducted in Portugal only fourth and fifth year students were included. Moreover, its sample size was smaller than ours. On the other hand, the finding of this study was lower than the one found in Republic of South Africa, which was 6.9% [10]. This mismatch might related to a time frame of suicidal attempt, which was life time prevalence.

In this study as cumulative grade point average increased in a unit, the odds of having suicidal ideation decreased by 70%. This result is supported by studies conducted in Serbia [42] Colombia [39], and Nepal [26]. The possible justification for this association could be the fact that when grade point average decreases, the student might consider suicidal ideation due to feeling of hopelessness, worthlessness, fear of failure and doubts about academic competence [43]. A frequent suicidal ideation has also been linked with depression, which in turn affects attention and concentration span and negatively impacts the ability to learn, understand, and solve academic problems [44].

The alcohol-user participants were 2.26 times more likely to have suicidal ideation as compared to their counterparts. This finding is congruous with the study result from Portugal [40],

**Table 3. Factors associated with suicidal ideation and attempt among undergraduate medical students of Haramaya University, Harar, Ethiopia, 2019 (n = 710).**

| Characteristics | | Suicidal ideation | | COR, (95%CI) | AOR, (95%CI) | Suicidal attempt | | COR(95% CI) | AOR(95% CI) |
|---|---|---|---|---|---|---|---|---|---|
| | | Yes(N) | No(N) | | | Yes (N) | No(N) | | |
| Grade point average | | Mean 2.99 | SD ±0.43 | 0.33, (0.22–0.50) | 0.30, (0.18–0.49) ** | | | | |
| Sex | Male | 106 | 383 | 1 | 1 | 15 | 474 | 1 | 1 |
| | Female | 62 | 159 | 1.41, (0.98–2.03) | 1.31, (0.85–2.03) | 13 | 208 | 1.98, (0.92–4.23) | 1.88, (0.84–4.18) |
| Family history of suicidal attempt | Yes | 20 | 32 | 2.15, (1.20–3.90) | 1.41, (0.62–3.17) | 5 | 47 | 2.94, (1.07–8.08) | 1.69, (0.53–5.37) |
| | No | 148 | 510 | 1 | 1 | 23 | 635 | 1 | 1 |
| Family history of committed suicide | Yes | 11 | 21 | 1.74, (0.82–3.68) | 1.92, (0.77–4.82) | 3 | 29 | 2.70, (0.77–9.45) | 2.05, (0.49–8.63) |
| | No | 157 | 521 | 1 | 1 | 25 | 653 | 1 | 1 |
| Family history of mental illness | Yes | 22 | 46 | 1.63, (0.95–2.79) | 0.91, (0.43–1.93) | | | | |
| | No | 146 | 496 | 1 | 1 | | | | |
| Current alcohol use | Yes | 78 | 120 | 3.05, (2.12–4.39) | 2.26, (1.45–3.55) ** | 12 | 186 | 2.00, (0.93–4.31) | 1.23, (0.51–3.51) |
| | No | 90 | 422 | 1 | 1 | 16 | 496 | 1 | 1 |
| Current khat use | Yes | 51 | 88 | 2.25, (1.51–3.36) | 1.39, (0.84–2.30) | 9 | 130 | 2.01, (0.89–4.55) | 1.34, (0.51–3.51) |
| | No | 117 | 454 | 1 | | 19 | 552 | 1 | 1 |
| Depression | Yes | 131 | 193 | 6.30, (4.27–9.60) | 3.58, (2.23–5.76) ** | 25 | 299 | 10.7, (3.19–35.69) | 5.40, (1.45–20.14) ** |
| | No | 37 | 349 | 1 | 1 | 3 | 383 | 1 | 1 |
| Anxiety | Yes | 130 | 204 | 5.67, (3.80–8.47) | 3.00, (1.88–4.77) ** | 24 | 310 | 7.20, (2.47–20.97) | 3.19, (1.01–10.18) * |
| | No | 38 | 338 | 1 | 1 | 4 | 372 | 1 | 1 |
| Stress | Yes | 93 | 229 | 1.70, (1.20–2.40) | 1.29, (0.85–1.96) | 18 | 304 | 2.24, (1.02–4.92) | 1.46, (0.64–3.36) |
| | No | 75 | 313 | 1 | 1 | 10 | 378 | 1 | 1 |
| Social support | Poor | 89 | 160 | 3.24, (1.92–5.45) | 2.57, (1.41–4.68) * | 17 | 232 | 2.13, (0.77–5.88) | 1.25, (0.54–4.41) |
| | Moderate | 57 | 254 | 1.31, (0.76–2.23) | 1.13, (0.62–2.09) | 6 | 305 | 0.57, (0.17–1.90) | 0.43, (0.12–1.49) |
| | Strong | 22 | 128 | 1 | 1 | 5 | 145 | 1 | 1 |

* = p<0.05, and

** = p<0.001.

Pakistan [13], and Serbia [42]. This might be due to the fact that drinking alcohol often leads to increased impulsiveness, poor judgment and weak resistance to dangerous behaviors. They might also consider suicidal ideation in withdrawal state due to dysphoric feeling associated with alcohol withdrawal [45, 46].

The participants with poor social support were 2.57 times more likely to have suicidal ideation as compared to those with strong social support. This is congruent with the study results from Nepal [26] and Pakistan [13]. A feeling of being neglected by neighbors, parents, and significant others might lead to worthless feeling, which in turn leads to suicidal ideation [47].

Regarding depression, the participants who had depression were 3.58 and 5.40 times more likely to have suicidal ideation and attempt than their counter parts respectively. This is

supported by the study conducted in USA [48], Colombia [39], Brazil [49], Portugal [40], China [24], Nepal [25], and South Africa [10]. Many of these previous studies have proven that the presence of depression is highly associated with suicidal ideation and attempt. The possible reason might be an association between decreased level of serotonin and its metabolite 5-hydroxyindoleacetic acid in the brain of suicidal individuals [50]. It may also be due to a direct effect of depression which makes individuals to feel hopeless, isolated and worthless [28].

The students with anxiety were three times more likely to have suicidal ideation and attempt than the participants who had no anxiety. This is compatible with study reports from China [15] and Portugal [40]. Fear of adapting new environment, increased psychosocial stress and academic pressures might lead to anxiety and, this in turn, leads to suicidal ideation and attempt [7, 51–53].

## Limitation of study

Since the data were collected through self-administered technique, we were unable to link the students with suicidal behavior with psychiatric service. We were able to report only associations, not definitive temporal or causal relationships, between suicidal ideation, attempt and the significantly associated factors. A recall bias might have also incurred.

## Conclusion and recommendation

This study revealed that suicidal ideation and attempt were common among undergraduate medical students in Haramaya University, College of Health and Medical Sciences. The magnitude was higher as compared to the prevalence of suicidal ideation among other university students who were studying in other fields. Cumulative grade point average, current alcohol use, depression, anxiety and poor social support were statistically associated with suicidal ideation, whereas depression and anxiety were the factors associated with suicidal attempt.

These affects the psychological wellbeing, academic performance and overall life of the medical students. Early screening, detection and management of suicidal behavior and associated mental health problems are recommended for undergraduate medical students. We would also like researchers to conduct comparative and longitudinal study to identify the causal relationship between suicidal ideation and attempt.

## Supporting information

**S1 Dataset. Data set used for analysis.**
(SAV)

## Acknowledgments

We are grateful to data collectors for their admirable endeavor. Also our appreciation goes to study participants who willingly contributed to this study by responding to the questionnaires.

## Author Contributions

**Conceptualization:** Niguse Yigzaw, Zegeye Yohannis, Gelana Fekadu, Yadeta Alemayehu.

**Formal analysis:** Henock Asfaw.

**Methodology:** Henock Asfaw.

**Supervision:** Niguse Yigzaw, Zegeye Yohannis.

**Visualization:** Niguse Yigzaw, Zegeye Yohannis.

**Writing – original draft:** Henock Asfaw.

**Writing – review & editing:** Henock Asfaw, Gelana Fekadu, Yadeta Alemayehu.

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
