## [Decision Letter · Decision Letter 0]

28 Apr 2020

PONE-D-20-08024

Prevalence and associated factors of suicidal ideation and attempt among undergraduate medical students of Haramaya University, Ethiopia. A cross sectional study

PLOS ONE

Dear Mr Asfaw,

Thank you for submitting your manuscript to PLOS ONE. After careful consideration, we feel that it has merit but does not fully meet PLOS ONE’s publication criteria as it currently stands. Therefore, we invite you to submit a revised version of the manuscript that addresses the points raised during the review process.

We would appreciate receiving your revised manuscript by Jun 12 2020 11:59PM. To enhance the reproducibility of your results, we recommend that if applicable you deposit your laboratory protocols in protocols.io, where a protocol can be assigned its own identifier (DOI) such that it can be cited independently in the future. For instructions see: http://journals.plos.org/plosone/s/submission-guidelines#loc-laboratory-protocols

We look forward to receiving your revised manuscript.

Kind regards,

Marco Innamorati

Academic Editor

PLOS ONE

Journal Requirements:

2.  We noticed you have some minor occurrence of overlapping text with the following previous work, which needs to be addressed:

https://doi.org/10.1016/j.jflm.2014.08.006

In your revision ensure you cite all your sources (including your own works), and quote or rephrase any duplicated text outside the methods section. Further consideration is dependent on these concerns being addressed.

3. Please provide additional details regarding participant consent. In the Methods section, please state why it was not possible to obtain written consent, how verbal consent was recorded and whether the ethics committee approved this consent procedure. If your study included minors, state whether you obtained consent from parents or guardians.

6. Please ensure that you refer to Figure 1 in your text as, if accepted, production will need this reference to link the reader to the figure.

7. Please include your tables as part of your main manuscript and remove the individual files. Please note that supplementary tables (should remain/ be uploaded) as separate "supporting information" files.

Additional Editor Comments (if provided):

Reviewers' comments:

Reviewer's Responses to Questions

**Comments to the Author**

1. Is the manuscript technically sound, and do the data support the conclusions?

Reviewer #1: Yes

Reviewer #2: Partly

2. Has the statistical analysis been performed appropriately and rigorously? 

Reviewer #1: Yes

Reviewer #2: Yes

3. Have the authors made all data underlying the findings in their manuscript fully available?

Reviewer #1: Yes

Reviewer #2: Yes

4. Is the manuscript presented in an intelligible fashion and written in standard English?

Reviewer #1: Yes

Reviewer #2: Yes

5. Review Comments to the Author

Reviewer #1: This is, in summary, an interesting paper aimed to assess the prevalence and associated factors of suicidal ideations and attempts among undergraduate medical students of Haramaya University in a sample of 757 subjects. The study showed that the prevalence of suicidal ideation and attempts were 23.7% and 3.9%, respectively. Cumulative Grade Point Average, current alcohol use, depression, anxiety, and poor social support were significantly associated with suicidal ideation. In addition, depression and anxiety were linked to suicidal attempts.

The authors may find as follows my main comments/suggestions.

First, when within the Introduction section, the authors correctly reported that suicial behavior is a major health priority being the second leading cause of death among adolescents and young adults, they might even briefly refer to the possible link between suicidal behavior, biological abnormalities, and emotional turmoil related to bereavement. Specifically, the emotional turmoil in suicide survivors of patients died by suicide may last a long time, and in some cases, may end with their own suicide. Thus, it is fundamental to understand the bereavement process after the suicide of a significant other in order to provide a proper care, and improve the outcomes. In order to briefly discuss this topic (although i understand that the link between suicidal behavior, and emotional turmoil related to bereavement process is not the main topic of the present manuscript), i suggest to discuss and cite, within the main text, the following papers (PMID: 31647957, 24082246, 30601750). In addition, specific biological abnormalities (e.g., prolactin, thyroid hormones, and immune alterations) may be significantly associated with suicidal behavior even involved in a complex compensatory mechanism to correct the abnormal central serotonin activity. The assumption that specific biological abnormalities may be associated or even predict suicidal behavior is of great importance, given the availability of such data in everyday clinical practice. Physicians of any kind as well as mental health professionals should be aware of the importance to insert as much information possible in the assessment of suicide. Thus, given the above information, my additional suggestion is also to briefly discuss and include throughout the manuscript the following additional reports (PMID: 30601750, 22748186, 29926090).

Moreover, as the main aims/objectives of this paper have been well described, the most relevant hypotheses underlying the present study should be reported as well in a detailed manner.

Furthermore, the section results is, in my opinion, divided in too many subsections that are difficult to follow for the general readership. Here, the most relevant study findings need to be better summarized for clarity.

In addition, the section limitations/shortcomings needs to be more directly updated as the most relevant caveats have been only partially described.

Also, the manuscript needs to be reviewed by a native English speaker for the quality of language.

Finally, what is the take-home message of this study? While the authors reported that the university should give due emphasis to reduce the risk factors and provide psychosocial support, they failed to report the most relevant conclusive remarks of their paper. Here, more details/information are required.

Reviewer #2: This is an interesting study with a potential clinical relevance focusing on suicidal behaviour in undergraduate medical students. The paper present an important topic, with a concise introduction, adequate and well described methodology, well presented results and conclusions that are supported by the findings. A few comments to improve the paper:

1. The language of the paper should be tuned to be a bit more "scientific" please work on this. Also correct several typing mistakes, starting with capitalisation of affiliation, missing or unnecessary spaces, etc.

2. Abstract: the same applies. I believe also that if you provide a definition of suicide as PLOS has a general audience please keep it less "naive". Also provide more sound rationale in the abstract for this study. Also: conclusion of abstract should express more than prevalence was "high". compared to what?

3. Methods: there was a one-month data collection period. Suicidal behaviour is linked to several stress- and environment related factors. Why I am not familiar with possible year-round fluctuations of Ethiopian weather, which is present in more northern home countries, the lack of such influencing factors should also be mentioned. Furthermore, i wonder how this period related to other possible risk stress factors like onset exam period. Therefore please comment on the choice of this period for data collection and the possible associated factors which may have to be considered when interpreting the data.

4. Was the sample representative with respect to factors other than year of medical university? such as socioeconomic background, family background, age, religion, etc?

5. How was suicidal ideation specifically assessed, by what question?

6. Was there correction for possible confounding variables?

7. Were students with psychiatric disorders excluded? Was there screening for depression or anxiety and correction for these variables? Is there data on medications taken that could influence depression and suicidality in any direction?

8. Is there data available in suicidal behaviour prevalence in Ethiopia? It would be necessary to compare the prevalence found in this study to genera population data.

In general although I have a few problems with the present study, the findings are interesting and the discussion is, while fully supported by the data, complex and thorough. After major revision the paper is worthy of publication.

6. PLOS authors have the option to publish the peer review history of their article (what does this mean?). If published, this will include your full peer review and any attached files.

Reviewer #1: No

Reviewer #2: No

---

## [Author Response · Author response to Decision Letter 0]

13 Jun 2020

Dear Reviewers, 

It is my great pleasure to respond to your crucial and constructive comments. It helped me to know how much the scientific writing needs attention, specificity and clarity. I learned much as a junior researcher. Herewith I have addressed all comments and suggestions accordingly. Thanks once again.

Author,

Henock Asfaw

Response: Thank you. I have corrected the manuscript according to the Plos one journal guideline. 

2. We noticed you have some minor occurrence of overlapping text with the following previous work, which needs to be addressed: https://doi.org/10.1016/j.jflm.2014.08.006

Response: Thank you. I have addressed it.

In your revision ensure you cite all your sources (including your own works), and quote or rephrase any duplicated text outside the methods section. Further consideration is dependent on these concerns being addressed.

3. Please provide additional details regarding participant consent. In the Methods section, please state why it was not possible to obtain written consent, how verbal consent was recorded and whether the ethics committee approved this consent procedure. If your study included minors, state whether you obtained consent from parents or guardians.

Response: Thank you. The ethical consideration included under the method section.

Ethical consideration

Ethical approval was obtained from a joint ethical review committee of the University of Gondar, College of medicine and health sciences and Amanuel mental specialized hospital. Formal letter was obtained from Amanuel mental specialized hospital and submitted to Haramaya University, college of health and medical science for administrative approval prior to the data collection. Informed written consent was obtained from the students prior to the data collection. Those who did not wish to take part could be allowed either to withdraw from study at any time they want. 

Response: Thank you. I have secured the ORCID ID.

Response: Thank you. I will consider it and make it identical.

6. Please ensure that you refer to Figure 1 in your text as, if accepted, production will need this reference to link the reader to the figure.

Response: Thank you. Figure 1 included as reference under sample size and sampling technique section. It is provided as a separate file.

7. Please include your tables as part of your main manuscript and remove the individual files. Please note that supplementary tables (should remain/ be uploaded) as separate "supporting information" files.

Response: thank you. I have incorporated all tables in the manuscript. 

Additional Editor Comments (if provided):

Reviewers' comments:

Reviewer's Responses to Questions

Comments to the Author

1. Is the manuscript technically sound, and do the data support the conclusions?

Reviewer #1: Yes

Reviewer #2: Partly

2. Has the statistical analysis been performed appropriately and rigorously?

Reviewer #1: Yes

Reviewer #2: Yes

3. Have the authors made all data underlying the findings in their manuscript fully available?

Reviewer #1: Yes

Reviewer #2: Yes

4. Is the manuscript presented in an intelligible fashion and written in Standard English?

Reviewer #1: Yes

Reviewer #2: Yes

5. Review Comments to the Author

Reviewer #1: 

Comment: This is, in summary, an interesting paper aimed to assess the prevalence and associated factors of suicidal ideations and attempts among undergraduate medical students of Haramaya University in a sample of 757 subjects. The study showed that the prevalence of suicidal ideation and attempts were 23.7% and 3.9%, respectively. Cumulative Grade Point Average, current alcohol use, depression, anxiety, and poor social support were significantly associated with suicidal ideation. In addition, depression and anxiety were linked to suicidal attempts.

The authors may find as follows my main comments/suggestions.

First, when within the Introduction section, the authors correctly reported that suicidal behavior is a major health priority being the second leading cause of death among adolescents and young adults, they might even briefly refer to the possible link between suicidal behavior, biological abnormalities, and emotional turmoil related to bereavement. Specifically, the emotional turmoil in suicide survivors of patients died by suicide may last a long time, and in some cases, may end with their own suicide. Thus, it is fundamental to understand the bereavement process after the suicide of a significant other in order to provide a proper care, and improve the outcomes. In order to briefly discuss this topic (although I understand that the link between suicidal behavior, and emotional turmoil related to bereavement process is not the main topic of the present manuscript), I suggest to discuss and cite, within the main text, the following papers (PMID: 31647957, 24082246, and 30601750). 

In addition, specific biological abnormalities (e.g., prolactin, thyroid hormones, and immune alterations) may be significantly associated with suicidal behavior even involved in a complex compensatory mechanism to correct the abnormal central serotonin activity. The assumption that specific biological abnormalities may be associated or even predict suicidal behavior is of great importance, given the availability of such data in everyday clinical practice. Physicians of any kind as well as mental health professionals should be aware of the importance to insert as much information possible in the assessment of suicide. Thus, given the above information, my additional suggestion is also to briefly discuss and include throughout the manuscript the following additional reports (PMID: 30601750, 22748186, 29926090).

Moreover, as the main aims/objectives of this paper have been well described, the most relevant hypotheses underlying the present study should be reported as well in a detailed manner.

Furthermore, the section results is, in my opinion, divided in too many subsections that are difficult to follow for the general readership. Here, the most relevant study findings need to be better summarized for clarity.

In addition, the section limitations/shortcomings needs to be more directly updated as the most relevant caveats have been only partially described.

Also, the manuscript needs to be reviewed by a native English speaker for the quality of language.

Finally, what is the take-home message of this study? While the authors reported that the university should give due emphasis to reduce the risk factors and provide psychosocial support, they failed to report the most relevant conclusive remarks of their paper. Here, more details/information are required.

Response: Thank you. It is great recommendation to include the biological abnormalities related to suicidal behavior. The recommended articles helped me to get a better insight. I have incorporated briefly about the prolactin and thyroid hormone alterations linked to suicidal behavior. Introduction section. Para 7, line VII

I have considered the take- home message and have modified the conclusion part as follows”…. Cumulative grade point average, current alcohol use, depression, anxiety and poor social support were statistically associated with suicidal ideation, whereas Depression and anxiety were factors associated with suicidal attempts.” Para 1, line 3 to 5

Reviewer #2: 

This is an interesting study with a potential clinical relevance focusing on suicidal behavior in undergraduate medical students. The paper present an important topic, with a concise introduction, adequate and well described methodology, well presented results and conclusions that are supported by the findings. A few comments to improve the paper:

1. The language of the paper should be tuned to be a bit more "scientific" please work on this. Also correct several typing mistakes, starting with capitalization of affiliation, missing or unnecessary spaces, etc.

Response: Thank you. The language have edited by native English speaker. 

2. Abstract: the same applies. I believe also that if you provide a definition of suicide as PLOS has a general audience please keep it less "naive". Also provide more sound rationale in the abstract for this study. Also: conclusion of abstract should express more than prevalence was "high". Compared to what?

Response: Thank you. The first line of abstract section have modified. Para I. line 1.

 Under the conclusion of abstract, I have re-written the conclusion by incorporating the study findings precisely. Conclusion: Para I, line 2 to 5.

3. Methods: there was a one-month data collection period. Suicidal behavior is linked to several stress- and environment related factors. Why I am not familiar with possible year-round fluctuations of Ethiopian weather, which is present in more northern home countries, the lack of such influencing factors should also be mentioned. Furthermore, I wonder how this period related to other possible risk stress factors like onset exam period. Therefore please comment on the choice of this period for data collection and the possible associated factors which may have to be considered when interpreting the data.

Response: Thank you. I do share your concern about the effect of stress and environment related factor on suicidal ideation and attempt. The weather in Ethiopia during May and June is the time with very favorable weather condition. The May is the spring season with occasional rain and hot weather while the June is summer season with a rain falls. So, by considering the weather situation we have collected the data from May 13 to June 12, which was the most comfortable and ideal with moderate hot and slight rainy weather. Concerning the exam, the data collection time was conducted during the exam free time (mid exam and final exam). The study participants were attending the regular class at that time. 

4. Was the sample representative with respect to factors other than year of medical university? Such as socioeconomic background, family background, age, religion, etc?

Response: Thank you. The reason behind taking the students’ year of medical university was by assuming the other characteristics as normally distributed among classes. The participants were selected for the study by using simple random sampling method, so it gave them equal opportunity to be the part of study. After the random selection the subjects were interviewed for characteristics such as socioeconomic background, family background, age, religion etc.

5. How was suicidal ideation specifically assessed, by what question?

Response: Thank you. Suicidal ideation were assessed by “Yes” or “No” question coded as 1 for “Yes” and 0 for “No”. The items were adapted from module of world mental health survey initiative version of the WHO, CIDI (composite international diagnostic interview), in which suicide is studied and validated in Ethiopia both at clinical and community settings. Its internal consistence (cronbach alpha) in current study is 0.85. Data collection tool and procedure section. Data collection tool and procedure, Para III, line 1 to 4.

6. Was there correction for possible confounding variables?

Response: Thank you. As it is known the major aim of epidemiological studies are to identify risk factors of disease based on association. There will be also another factor associated with the exposure of disease and distort the exposure outcome relationship. We call such factors a confounding factor or variable. Such variable could be managed at study design phase and data analysis phase. So, the effect of confounding variable could be excluded from the final result. At the design phase we can use randomization, restriction and matching technique. At the analysis phase we can use stratification and multivariate models. 

In current study we have used randomization technique. The study participants were selected randomly. Additionally during the analysis multivariate with logistic regression (bivariate and multivariate) analysis was conducted. 

7. Were students with psychiatric disorders excluded? Was there screening for depression or anxiety and correction for these variables? Is there data on medications taken that could influence depression and suicidality in any direction?

Response: Thank you. Unfortunately there was no student with known psychiatric disorder among the study subjects according to the data we have got from the school of medicine head. Concerning the depression or anxiety, we have included the questions to screen them. Depression, anxiety and stress was collected by DASS, which have twenty one items and seven items each and used to assess emotional state of depression, anxiety and stress. 

Data collection tool and procedure, Para I, line 6 to 9.

The information related to medication/substance use also included in our tool. ASSIST (Alcohol, Khat, Cigarette and Others) were used to collect data related to psychoactive substances which were developed by WHO. Data collection tool and procedure, Para I, line 4 to 5.

8. Is there data available in suicidal behavior prevalence in Ethiopia? It would be necessary to compare the prevalence found in this study to genera population data.

Response: Thank you. I have incorporated the prevalence of suicidal behavior among general population in Ethiopia. Introduction. Para VI, line 4 to 5. 

In general although I have a few problems with the present study, the findings are interesting and the discussion is, while fully supported by the data, complex and thorough. After major revision the paper is worthy of publication.

6. PLOS authors have the option to publish the peer review history of their article (what does this mean?). If published, this will include your full peer review and any attached files.

Response: Thank you. Am grateful to publish my future works on PloS one.

---

## [Decision Letter · Decision Letter 1]

8 Jul 2020

Prevalence and associated factors of suicidal ideation and attempt among undergraduate medical students of Haramaya University, Ethiopia. A cross sectional study

PONE-D-20-08024R1

Dear Dr. Asfaw,

We’re pleased to inform you that your manuscript has been judged scientifically suitable for publication and will be formally accepted for publication once it meets all outstanding technical requirements.

Kind regards,

Marco Innamorati

Academic Editor

PLOS ONE

Reviewers' comments:

Reviewer's Responses to Questions

**Comments to the Author**

1. If the authors have adequately addressed your comments raised in a previous round of review and you feel that this manuscript is now acceptable for publication, you may indicate that here to bypass the “Comments to the Author” section, enter your conflict of interest statement in the “Confidential to Editor” section, and submit your "Accept" recommendation.

Reviewer #1: All comments have been addressed

Reviewer #2: All comments have been addressed

2. Is the manuscript technically sound, and do the data support the conclusions?

Reviewer #1: Yes

Reviewer #2: Yes

3. Has the statistical analysis been performed appropriately and rigorously? 

Reviewer #1: Yes

Reviewer #2: Yes

4. Have the authors made all data underlying the findings in their manuscript fully available?

Reviewer #1: Yes

Reviewer #2: Yes

5. Is the manuscript presented in an intelligible fashion and written in standard English?

Reviewer #1: Yes

Reviewer #2: Yes

6. Review Comments to the Author

Reviewer #1: In the revised manuscript, the authors addressed successfully most of the major questions raised by Reviewers improving both the main structure and quality of the present paper. I have no further additional comments.

Reviewer #2: The authors ahve addressed all comments raised by the reviewer whch improved this already good paper. Therefore I recommend to publish the paper in its present form. I also congratulate the authors, especially give that they arejunior scientists.

7. PLOS authors have the option to publish the peer review history of their article (what does this mean?). If published, this will include your full peer review and any attached files.

Reviewer #1: No

Reviewer #2: No

---

## [Editor Report · Acceptance letter]

21 Jul 2020

PONE-D-20-08024R1 

Prevalence and associated factors of suicidal ideation and attempt among undergraduate medical students of Haramaya University, Ethiopia. A cross sectional study 

Dear Dr. Asfaw:

I'm pleased to inform you that your manuscript has been deemed suitable for publication in PLOS ONE. Congratulations! Your manuscript is now with our production department. 

Kind regards, 

on behalf of

Dr. Marco Innamorati 

Academic Editor

PLOS ONE